# Role of Adiponectin Peptide I (APNp1) in Age-Related Macular Degeneration

**DOI:** 10.3390/biom12091232

**Published:** 2022-09-03

**Authors:** Connor Logan, Valeriy Lyzogubov, Nalini Bora, Puran Bora

**Affiliations:** Pat & Willard Walker Eye Research Center, Jones Eye Institute, Department of Ophthalmology, University of Arkansas for Medical Sciences, 4301 West Markham, Little Rock, AR 72205, USA

**Keywords:** age related macular degeneration, adiponectin, adiponectin receptor, neovascularization, adeno-associated virus, topical administration

## Abstract

Age-related macular degeneration (AMD) is an eye disease that can cause central vision loss, particularly in the elderly population. There are 2 classes of AMD, wet-type and dry-type. Wet-type involves excess angiogenesis around the macula, referred to as choroidal neovascularization (CNV). This can result in leaky vessels, often causing more severe vision loss than dry-type AMD. Adiponectin peptide 1 (APNp1) has been shown to slow the progression of CNV. Here, we used a mouse model and FITC-labeled APNp1 to determine if APNp1 could be delivered effectively as an eye drop. Our experiment revealed that topically applied FITC-APNp1 could reach the macula of the eye, which is crucial for treating wet-type AMD. We also tested delivery of APNp1 via injection of an adeno-associated virus (AAV) vector in a mouse model of CNV. AAV is a harmless virus easy to manipulate and is very often used for protein or peptide deliveries. Results revealed an increase in the expression of APNp1 in the retina and choroid over a 28-day period. Finally, we investigated the mechanism by which APNp1 affects CNV by examining the expression of adiponectin receptor 1 (AdipoR1) and proliferating cell nuclear antigen (PCNA) in the retinal and choroidal tissue of the mouse eyes. AdipoR1 and PCNA were overexpressed in these tissues in mice with laser-induced CNV compared to naïve mice. Based on our data shown here, we think it will enhance our understanding of APNp1 as a therapeutic agent for wet-type AMD and possible treatment alternatives that could be more beneficial for patients.

## 1. Introduction

### 1.1. Age-Related Macular Degeneration

Age-related macular degeneration (AMD) is a major cause of central vision loss in individuals over 50 years old, and there are more than 4 million people in the United States alone who have AMD [1,2,3,4,5,6,7,8,9,10,11]. There are 2 primary clinical forms of AMD: dry-type (non-exudative) and wet-type (exudative). Dry-type is more prevalent than wet-type, but wet-type often causes more severe vision loss due to complications of choroidal neovascularization (CNV) [1,2,3,4,5,6,7,8,9,10,11]. The progression of early AMD to late AMD (geographic atrophy and neovascular AMD) can have serious consequences, including central vision loss, difficulty to read and recognizing faces and objects. This can affect quality of life and lead to depression [1,2,3,4]. Because the human lifespan is increasing, the number of patients with AMD is increasing [2,3,4,5]. Currently, it costs an estimated $900 million in the United States annually to treat AMD [1,2,3,5,6].

There are treatments available for wet AMD, Genentech developed humanized antibody against VEGF Avastin and Lucentis, both are equally good. Avastin was first used only for cancer treatment, but now it is also used for wet AMD treatment. However, Lucentis is 20 times more expensive than Avastin. Regeneron Pharmaceuticals developed Eylea for wet AMD treatment, but Eylea is as expensive as Lucentis. These drugs tend to be expensive and come with serious side effects, including infection, hemorrhage, and ocular pain [9,10,11,12,13]. Wet AMD occurs when CNV network damages the macula and the central vision stops working. CNV distorts the structure of the macula and disrupts the function of rods and cones of the photoreceptors located in that area. This damages the cells and microenvironment of the macula leads to scarring and potentially rapid central vision loss. CNV itself is a complex biological process and its underlying molecular mechanisms are still unknown [13,14,15,16,17,18,19]. Understanding such mechanisms will lead to better characterization of CNV and can help researchers identify new molecular targets (such as adiponectin receptor 1) to develop therapeutic strategies.

### 1.2. Adiponectin Peptide 1 (APNp1)

Adiponectin (APN) is a pleiotropic protein responsible for regulating cellular functions such as cell proliferation [16,20,21,22,23,24,25,26,27,28,29,30,31,32,33,34,35,36,37,38,39,40,41,42]. APN is produced by both adipocytes and importantly by choroidal [16,21,22,23,24,25,26,27,28,29,30,31,32]. This led us to investigate the potential role of adiponectin peptide 1 (APNp1) in the treatment of CNV. APNp1 is an anti-angiogenic peptide with 18 amino acids and is an internal fragment of adiponectin. We found that APNp1 inhibited choroidal endothelial cell (CEC) proliferation and inhibited CNV by more than 75% in mouse and rat models [16,20]. It was originally thought that APNp1 inhibited CNV by inhibiting CEC proliferation; however, we discovered that APNp1 also affects the retinal pigmented epithelium (RPE) by promoting RPE proliferation and migration. Both of these effects of APNp1 are required for the effective treatment of wet AMD. Thus, APNp1 may be the first potential multifaceted wet-type AMD therapeutic agent, affecting both CECs and the RPE. Current treatment modalities for wet AMD only target CECs. The possible mechanism of action of APNp1 on CECs and RPE cells will be discussed here.

### 1.3. Topical Therapeutic Administration

We have established that APNp1 inhibits CNV [16,20]. Current treatments for wet AMD, such as VEGF inhibitors, typically require repeated injections into the eye and come with major limitations. The efficacy of ocular injections significantly decreases over time [43,44,45], and there are risks of retinal detachment, infection, hemorrhage, and ocular pain [8,9,10,13]. As such, a less invasive technique, such as topical administration, would be very beneficial to patients undergoing AMD treatment. A major advantage of topical administration is that patients can use eye drops by themselves. This reduces the need for frequent clinic visits for injections, thus reducing the time and financial load of treatment. Here, we investigated the ability of APNp1 to reach the posterior segment of the eye via eye drop.

### 1.4. Adeno-Associated Virus Therapeutic Delivery

The adeno-associated virus (AAV) vector delivery system can be a better option than intravitreal or subretinal injection for long-term expression of therapeutic gene(s) in the treatment of AMD. This is due to the extended period of gene expression from episomally stable DNA delivered by the vector. Here, we investigated the use of an AAV system to administer APNp1 in a mouse model of wet-type AMD. This therapeutic approach would still require ocular injections for patients, but the injections would be much less frequent. Thus, it would save patients time and money and decrease the risk of adverse effects of frequent injections. Our data provide new information regarding the potential use of AAV-delivered APNp1 to treat CNV (and wet AMD).

## 2. Materials and Methods

**Animals**: Male C57BL/B6 mice (6–8 weeks old) were purchased from The Jackson Laboratory (Bar Harbor, ME, USA). This study was approved (ID# Puran Bora 4008) by the Institutional Animal Care and Use Committee at the University of Arkansas for Medical Sciences, Little Rock, AR, USA.

Each experimental group had 5 mice and all the experiments were repeated three times. All mice were fed a regular diet and were 6–8 weeks old when used for experiments.

### 2.1. Colocalization of APNp1 and AdipoR1

The colocalization of APNp1 and adiponectin receptor 1 (AdipoR1) was examined in choroidal endothelial cells (CECs) of naïve mice and mice with laser-induced CNV. CNV was induced via Argon laser photocoagulation whereby three laser spots were placed in each eye close to the optic nerve (spot size—50 um; duration—0.05 s; power—260 mW). APNp1 tagged with FITC, and laser confocal microscopy were used to analyze cryosections of mouse eyes.

### 2.2. Expression of AdipoR1 and PCNA

Paraffin sections of retinal pigmented epithelium were obtained from naïve mice and mice with laser-induced CNV and stained with antibodies against AdipoR1 and proliferating cell nuclear antigen (PCNA) via double immunohistochemistry, goat polyclonal anti-PCNA antibody (Biovision, Mountain View, CA, USA), diluted 1:2000, rabbit anti-adiponectin R1 (from Phoenix, Pharmaceuticals, Belmont, CA and Abcam, Cambridge, MA, USA), diluted 1:400. Reverse transcriptase PCR was used to assess AdipoR1 mRNA expression in mouse RPE–choroid tissue. In addition, human ARPE-19 cells were cultured for 72 h, fixed, and double immunocytochemically stained for AdipoR1 and PCNA; nuclei were counterstained with DAPI. Laser confocal microscopy was used to assay AdipoR1 in PCNA-positive cells and PCNA-negative cells.

### 2.3. Choroidal Endothelial Cells (CECs) Culture

CECs were isolated from eyes as described [21] with modifications. Mice (*n* = 5) were sacrificed, and eyes were removed under sterile conditions. Eyes were treated with 70 °C ethanol for 30 s and washed with Dulbecco’s phosphate buffered saline (DPBS, Mediatech, Manassas, VA, USA) for 1 min (3 times). The eyes were dissected, and the anterior part of the eyes, lens and retina was removed. RPE-choroid was separated from sclera by scrapping and transferred to DPBS. RPE-choroid tissue was treated with 0.25% Trypsin (HyClone, Logan, UT, USA) for 30 min. RPE-choroid cells were incubated in selective medium MCDB-1 with fetal bovine serum and antibiotics (VEC technologies, Rensselaer, NY, USA) in 25 cm^2^ flasks under condition of 5% CO_2_ and 37 °C. After 2 passages cells were investigated for presence of endothelial cell markers CD31, von Willebrand Factor (vWF) and Isolectin IB4 (ILIB4). We used CD31 + Vwf + ILIB4 + endothelial cells, from 3rd to 5th passage (split ratio 1:3), 50 cells per 1 mm^2^ of surface area.

### 2.4. Real-Time Quantitative RT-PCR (RT-qPCR)

Total RNA was purified using the RNeasy mini kit (Qiagen, Valencia, CA, USA), and cDNA was synthesized using the iScript cDNA synthesis kit (Bio-Rad, Hercules, CA, USA) with 0.5 µg of total RNA according to the manufacturer’s recommendations. qPCR was performed with primers specific for mouse APN, and β-actin using iQ SYBR Green Supermix in an iQ5 real-time PCR detection system (Bio-Rad, Hercules, CA, USA). The primers were designed and ordered from Integrated DNA Technologies (Coralville, IA, USA). The primer sequences used were as follows: mouse APN, 5′-CGG TAT CCC ATT GTG ACC AG-3′ (forward) and 5′-CGC TCC TGT TCC TCT TAA TCC-3′ (reverse); and mouse β-actin, 5′-AAC CCT AAG GCC AAC CGT GAA A-3′ (forward) and 5′-AGG CAT ACA GGG ACA ACA CA-3′ (reverse). Blast tool was used, Primers correspond to mouse adiponectin mRNA. https://www.ncbi.nlm.nih.gov/tools/primer-blast/primertoo (accessed on 29 August 2022), NCBI Reference Sequence: NM_009605.5, FASTA. Pilot real-time RT-qPCR experiments were performed to determine the optimal condition for each primer. All real-time RT-qPCR experiments were performed in duplicate. The primer specificity of the amplification product was confirmed by melting curve analysis of the reaction products using SYBR Green as well as by visualization on ethidium bromide-stained agarose (1.5%) gels. The housekeeping gene β-actin was used as an internal control, and the gene-specific mRNA expression was normalized against β-actin expression. iQTM5 optical system software (Bio-Rad; version 2.0) was used to analyze real-time RT-qPCR data and derive threshold cycle (CT) values according to the manufacturer’s instructions. The ΔΔCT method was used to transform CT values into relative quantities with S.D. The same software was used to calculate the normalized expression of the gene of interest, using β-actin as reference gene, and the results were expressed as normalized fold expression.

### 2.5. Topical Administration of APNp1

APN is highly conserved protein and APNp1 is a conserve peptide with half-life of 8hrs. APNp1 was mixed in PBS. One drop of FITC-APNp1 (5 mg/mL in PBS) was administered topically to both eyes (one drop in each eye) of the mice (*n* = 5) 3 times per day for 7 days (1 drop = 10 µL = 50 µg FITC-APNp1). On the seventh day, the mice were euthanized, and eyes were processed for experiments. An equal amount of PBS was administered to the control group with the same scheduling (*n* = 5). APNp1 was designed from the globular region of Adiponectin and has more than 75% homology with complement protein C1q. APNp1 and FITC-APNp1 both were synthetized by Peptide Biochemical Research Inc., Seattle, WA, USA.

### 2.6. Adeno-Associated Virus (AAV) Delivery

Recombinant AAV containing an expression cassette of APNp1 was generated (pGPAAV/CMV/ss-myc-tag-APNp1) [46,47,48,49]. AAV vector without the APNp1 cassette (pGPAAV/CMV/ss-myc-tag) was used as a control. APNp1 was inserted next to Xho1 and in between Xho1 and TR. The vector was injected into mice intravitreally (2 µL AAV), and the eyes were lasered after injection to induce CNV. Mice were sacrificed on day 4, 14, or 28 after being lasered. Control mice received AAV lacking APNp1 and were sacrificed on day 28. Delivery of APNp1 was assayed by staining FITC-dextran–perfused RPE–choroid–sclera flat mounts with antibodies for markers of RPE cells. Markers were mouse monoclonal anti-cytokeratin 18 IgG1 (Affinity Bioreagents, Golden, CO, USA) and phallotoxin-Alexa Fluor (AF) 594 (Invitrogen, Valencia, CA, USA). Confocal laser microscopy was used to obtain multiple Z-stack images of flat mounts of CNV samples [16,17,18,19,20].

## 3. Results

### 3.1. Mechanism of Action of APNp1 on CECs and RPE

Primary choroidal endothelial cells were treated with FITC-APNp1 100 µg. We used laser confocal microscopy to analyze the interaction between FITC- APNp1 and AdipoR1 in primary CECs of mice (Figure 1). The goal was to determine if AdipoR1 was present in primary CEC’s and to determine if APNp1 co-localizes with AdipoR1 in laser-induced CNV. Our data supports that AdipoR1 is found in primary CEC’s and does co-localize with APNp1 in laser-induced CNV.

We investigated the mechanisms behind APNp1’s ability to slow CNV progression by studying expression of AdipoR1 and PCNA. RPE cells of naïve mice without CNV were compared with RPE cells of mice with laser-induced CNV. Paraffin sections of the mouse RPE cells were obtained for both groups and stained using antibodies against AdipoR1 and PCNA via double immunohistochemistry. We observed by immunohistochemistry that naïve mouse RPE cells did not express PCNA, and AdipoR1 expression was weak in the same cells (Figure 2). However, AdipoR1 and PCNA were both overexpressed in RPE cells located near areas of CNV. Reverse transcriptase PCR (RT-PCR) analysis of RPE–choroid tissue demonstrated an increase in AdipoR1 mRNA in mice with CNV compared to naïve mice (Figure 2). Thus, in mice with CNV, we observed higher amounts of PCNA and AdipoR1, indicating that the choroid is more sensitive to APNp1 in CNV. 

Human ARPE-19 cells were examined to determine if they expressed AdipoR1. The cells were fixed and double-stained for AdipoR1 (green) and PCNA (red). DAPI (blue) was used to counterstain nuclei. We found increased levels of AdipoR1 in PCNA-positive cells (pink) compared to PCNA-negative cells (blue) (Figure 3). The cells that were used were not induced with CNV. This data supports that AdipoR1 and PCNA are expressed in human ARPE-19 cells.

### 3.2. Topical Administration of APNp1

We investigated whether topically applied APNp1 could reach the posterior segment of mice eyes, where it could stimulate RPE proliferation and inhibit CEC growth to slow progression of CNV. FITC-APNp1 was observed in the retina, RPE, and choroid, indicating that topical APNp1 is capable of reaching the posterior segment of mouse eyes (Figure 4). Thus, topical administration of APNp1 could be a viable option for treatment of wet AMD, but more research will be needed to determine the efficacy of this modality.

### 3.3. Adeno-Associated Virus (AAV) Delivery of APNp1

Schematic diagrams of pGPAAV/CMV/ss-myc-tag-APNp1 serotype Y730 (A), and pGPAAV/CMV/ss-myc-tag serotype Y730 (B) are shown in Figure 5.

We found that intravitreal injection of an AAV vector containing APNp1 (pGPAAV/CMV/ss-myc-tag-APNp1) resulted in the expression of APNp1 in different layers of the retina and choroid in mice affected with CNV (Figure 6).

After injection, mice were lasered to induce CNV and then sacrificed on day 4, 14, or 28 post laser-photocoagulation. Low expression of APNp1 was observed on day 4 (Figure 6A). However, increased expression of APNp1 was observed on day 14 (Figure 6B), with much higher expression seen on day 28 (Figure 6C). No expression of APNp1 was observed on day 28 in control mice injected with the control vector (Figure 6D). This confirmed that the AAV vector method of application was possible and has potential to be used in treatment applications.

## 4. Discussion

There are two forms of age-related macular degeneration (AMD)—dry-type and wet-type. Dry-type is the more common type, but wet-type is associated with more severe vision loss. Wet-type AMD is caused by the overgrowth of blood vessels on the macula that exudate and inhibit the ability of the macula to function properly. The growth of these vessels is referred to as choroidal neovascularization (CNV). There is very little investigation that has been done on the effect of adiponectin and APNp1. However, investigators have used different inhibitors other than APNp1 to investigate wet and dry AMD [9,12,13,14,16,20].

APNp1 was designed from the globular region of adiponectin. The globular region of adiponectin, a functional region, binds to its receptors [22,25,26,41]. We showed that FITC-APNp1 could penetrate to the back of the eyes of mice when administered as a topical eye drop. However more studies are needed to characterize the inhibition of CNV by APNp1 to determine safety and effective dosing before using APNp1 therapeutically in humans. Current treatments for CNV require frequent eye injections; a topical application would greatly simplify the treatment for patients. Moving forward, we propose to test the efficacy of such eye drops. Specifically, we think it is appropriate to investigate whether the therapeutic efficacy of topical APNp1 is comparable to or superior to that of injected APNp1.

An adeno-associated virus (AAV) vector was tested for its ability to express APNp1 in desired tissues. We found that vector-encoded expression of APNp1 in mouse eyes increased over a 28-day period. This indicates that an AAV vector could be used to modulate the production of APNp1, thus making the peptide more available over an extended period of time; such a therapeutic approach would decrease the number and frequency of the eye injections required by patients. However, the efficacy and therapeutic dose of AAV encoding APNp1 in experimental CNV are still not well understood. We will further investigate into the effects of APNp1 on CNV when it is delivered via an AAV vector—specifically, the effect of intravitreal injection of pGPAAV/CMV/ss-myc-tag-APNp1 in mice with laser-induced CNV. We have established that APNp1 inhibits CNV by 75% compared to controls [20]. Therefore, long-term expression of APNp1 from an AAV should effectively inhibit CNV. AAV has been used by several investigators to deliver drugs to different parts of human body. Currently, very frequently AAV is being used for several drug deliveries because it is more economical and easier to handle compared to other virus deliveries and has very little safety concerns [46,47,48,49].

Laser confocal microscopy was used to confirm that APNp1 bound to AdipoR1 in CECs. Immunohistochemistry was then used on paraffin sections of mouse RPE cells to assay AdipoR1 levels. We found that AdipoR1 levels were low in cells not affected by CNV, and PCNA was not produced at all. However, both AdipoR1 and PCNA levels were high in RPE cells affected by CNV. RT-PCR was also used to assay AdipoR1 mRNA in RPE–choroid tissue, and we found that AdipoR1 expression increased in mice with CNV compared to controls. Co-localization experiments of human ARPE-19 cells showed more AdipoR1 staining in PCNA-positive cells than in PCNA-negative cells. All these observations collectively suggest that AdipoR1 plays an important role in the proliferation of RPE cells. We found increased expression of AdipoR1 and PCNA in the laser photocoagulation area. As mentioned above, increased expression of AdipoR1 and PCNA plays important role in the proliferation of RPE and choroidal endothelial cells. Adiponectin expression in RPE and choroid is very low. However, it is even lower in the choroid [20]. Therefore, additional APNp1 is required to inhibit CNV. We now know that APNp1 inhibits new vessel formation and proliferation. However, further investigations are needed on AdipoR1 and its effect on CNV progression and severity.

Earlier studies done by investigators have shown that adiponectin or adiponectin peptides can inhibit proliferation and angiogenesis in the pathogenesis of atherosclerosis and cancer cells [25,26,27,33,37]. It may be possible to modify AdipoR1 to be more sensitive to APNp1 in RPE cells affected by CNV, making increasing APNp1’s ability to stop unnecessary angiogenesis possible. A more sensitive version of AdipoR1 would promote increased binding of APNp1 and theoretically increase APNp1’s effect on CNV. This could be another potential approach to develop a therapeutic agent(s) for the treatment of CNV/wet-type AMD.

## Figures and Tables

**Figure 1 biomolecules-12-01232-f001:**
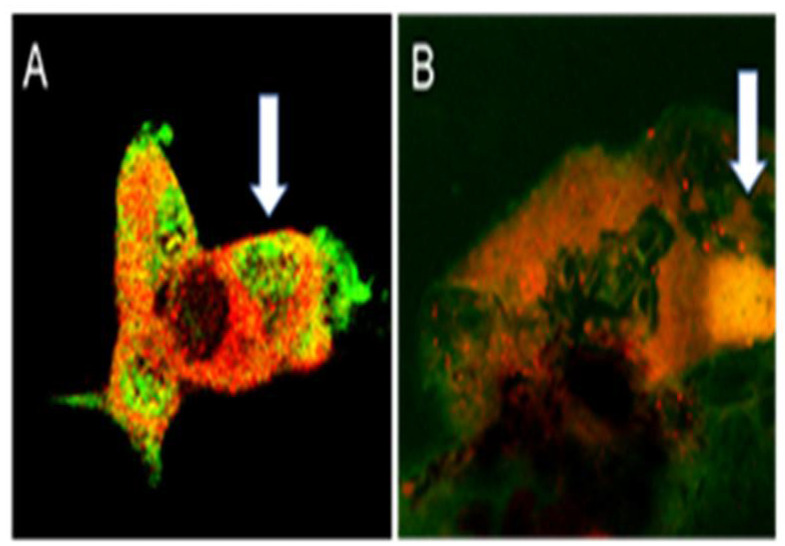
**APNp1 binding to AdipoR1**. (**A**) AdipoR1 (red) and FITC-APNp1 (green) in primary CEC. (**B**) AdipoR1 (red) expression and FITC-APNp1 (green) within laser-induced CNV. The arrow indicates colocalization (yellow).

**Figure 2 biomolecules-12-01232-f002:**
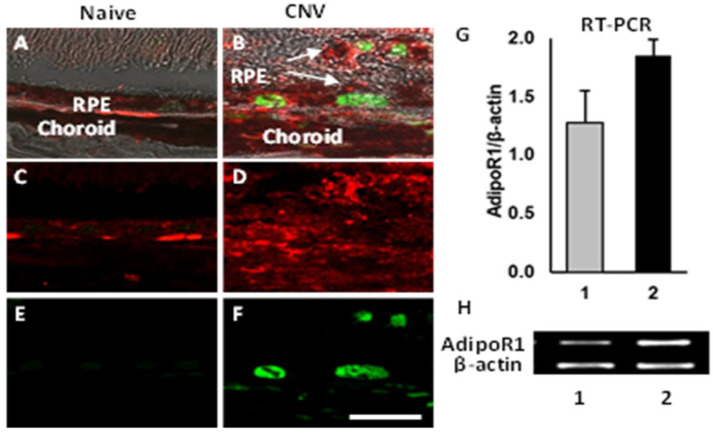
**Expression of AdipoR1 and PCNA in RPE cells.** Paraffin sections were stained using Ab against AdipoR1 and PCNA double IHC. Merged images of AdipoR1 (red color), PCNA (green color), and DIC (black and white) show colocalization of both markers and structures of the RPE-choroid complex (**A**,**B**). Red channel (**C**,**D**) and green channel (**E**,**F**) are shown separately. Panel (**G**) shows densitometric analysis of PCR products shown in Panel (**H**) for AdipoR1 and β-actin in RPE-choroid of naïve (lane 1) and laser-treated mice (lane 2) on day 7 after laser photocoagulation. Levels of AdipoR1 increased in laser treated animals. Bar for (**A**–**F**) = 20 µm.

**Figure 3 biomolecules-12-01232-f003:**
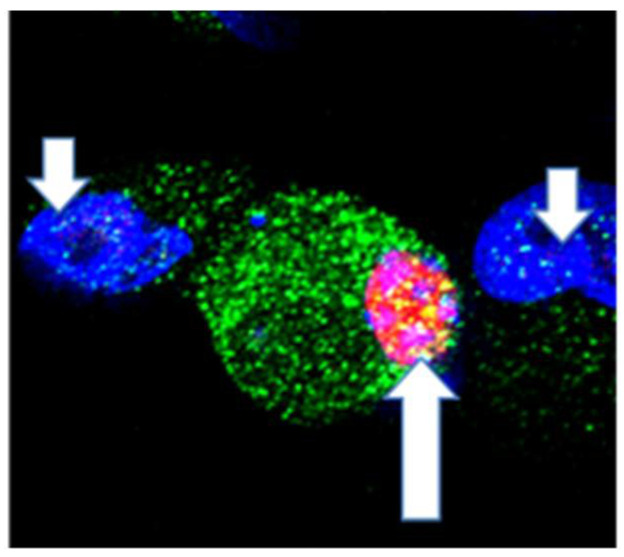
**Expression of AdipoR1 and PCNA in Human ARPE-19 cells.** Human ARPE-19 cells were fixed and double immunocytochemical staining for AdipoR1 (green color) and PCNA (red color) was performed. DAPI (blue color) was used for nuclei counterstaining. Laser confocal microscopy demonstrated increased levels of AdipoR1 in PCNA-positive cells in pink (indicated by upward arrow) compared to PCNA-negative in blue (indicated by downward arrow) cells. Magnification of objective is 40×.

**Figure 4 biomolecules-12-01232-f004:**
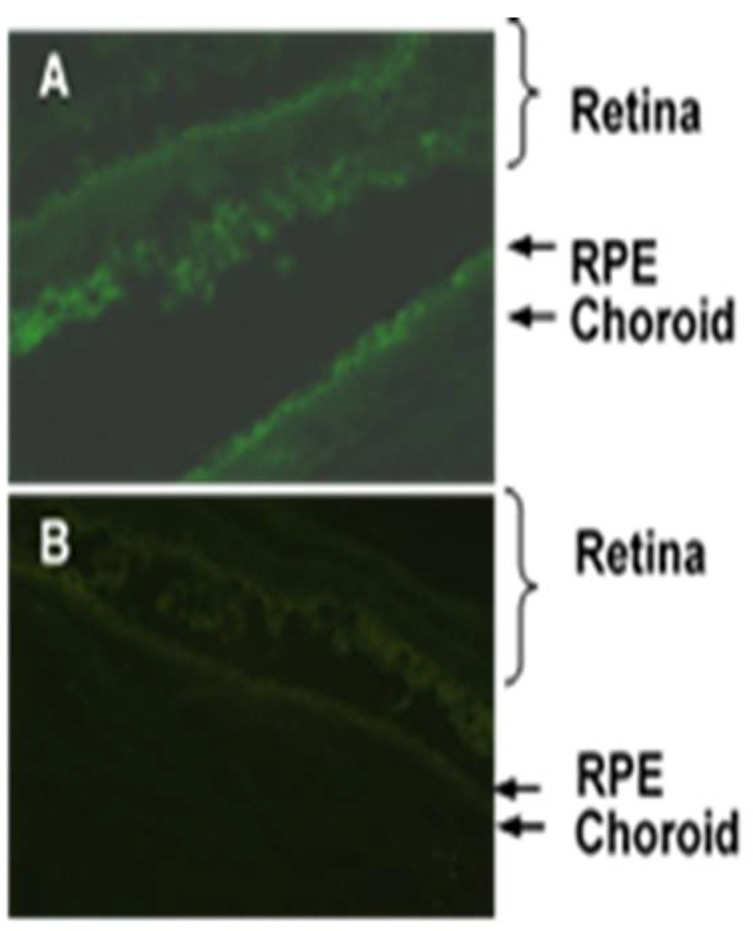
Topical Administration of FITC-APNp1 (**A**) or PBS (**B**) 3 times a day for 7 days. (**A**) Topical APNp1 reached the retina, RPE, and the choroid. (**B**) Weak flourescence was noted with the controls.

**Figure 5 biomolecules-12-01232-f005:**
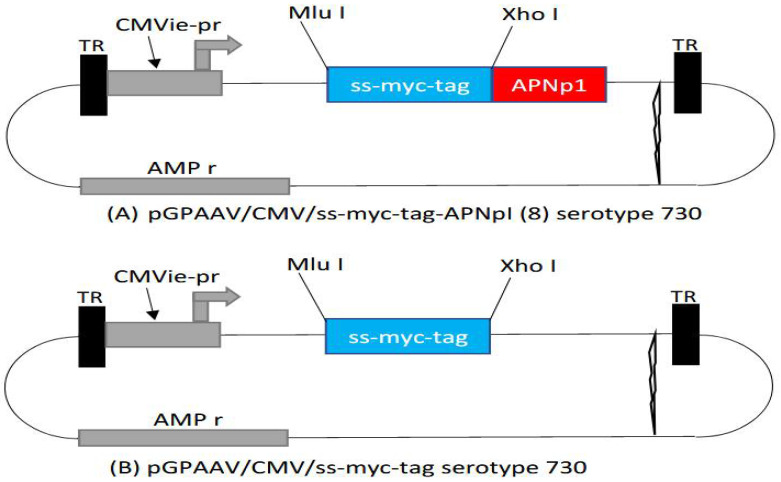
(**A**) pGPAAV/CMV/ss-myc-tag-APNpI (8) serotype 730; (**B**) pGPAAV/CMV/ss-myc-tag serotype 730. The top figure (**A**) is AAV with APNp1 inserted in it (Red) and the bottom figure (**B**) shows normal AAV with no APNp1 inserted in it, used as a control.

**Figure 6 biomolecules-12-01232-f006:**
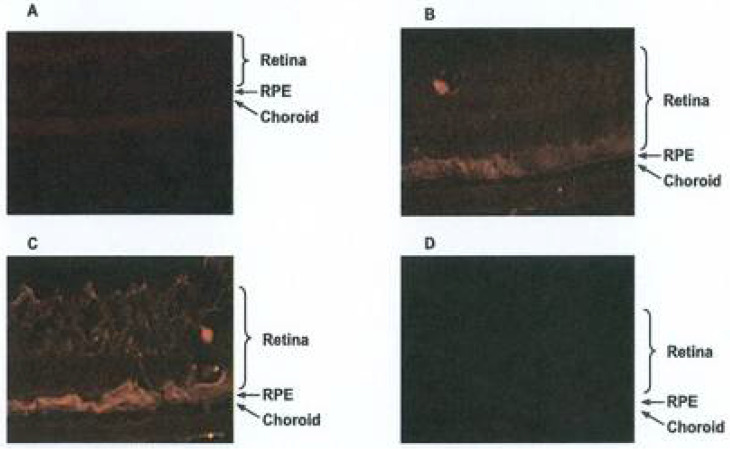
Mice were injected with 2 uL [2 × 10^6^ encapsidated genomes] of AAV encoding APNp1 and then lasered to induce CNV. Mice were sacrificed on days 4 (**A**), 14 (**B**), and 28 (**C**) post laser. Control mice were injected with AAV.Neo [2 × 10^6^ encapsidated genomes] and sacrificed on day 28 (**D**).

## Data Availability

Not applicable.

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
