# Peer review of "Role of Adiponectin Peptide I (APNp1) in Age-Related Macular Degeneration"

_biomolecules, 2022, doi:10.3390/biom12091232_

Round 1

Reviewer 1 Report

This manuscript by Logan et al., explores the possible role of APNp1 in age-related macular degeneration (AMD) in a mouse model of laser induced CNV as  well as in primary cultured CEC cells and RPE cell lines. More detailed information is needed in the methods section. I have some questions that need to be addressed.

1.    Antibodies are used in this study, the authors need to at least cite some references to mention the specificity of antibodies employed, concentration used, etc.

2.    The procedure for the immunofluorescence labeling in this manuscript is too sample, the authors need to add more detailed information to it, such as antibody was diluted in which buffer, blocking reagents used, antibody omission, washing buffer, incubation time. The laser confocal imaging needs to show whether the authors use single scan or Z-stack multiple scans, etc.

3.    The RT-PCR technique is used in this manuscript, however, it appears it is not qualify for the MIQE guidelines: minimum information for publication of quantitative real-time PCR experiments. The authors need to describe more detailed information, such as which reagents were used for RNA extraction, RNA quantification, PCR primers sequences, predicted PCR product length, no-template control showing negative results, etc.

4.    The authors need to give detailed information regarding how to isolate and primarily culture CEC cells, which medium was used?

5.    The quality of figure 6 is not good.

6.    Is “PBS” a better control than using “FITC” control in Figure 4.

7.    Is APNp1 peptide highly conserved, what is the half-life or turn over or degradation of APNp1 peptide?

8.    There are some additional issues, such as some figures need to add “scale bar”;  make “APNp1-FITC” and “FITC-APNp1” consistent. In Figure 6, please change 2x106 to 2X106.

Reviewer 2 Report

Role of Adiponectin Peptide I (APNp1) in Age-Related Macular Degeneration

Authors have investigated role of adiponectin peptide I and its topical administration as well as combined to AAV vector and intravitreally injected to mice eyes. Also its receptor localization upon laser induced CNV was investigated.

Title

Title is nice and easy to read because it is short but basicly it does not tell anything about results or conclusion. There could be added shortly some main conclusion based to the results e.g. Adiponectin Peptide I (APNp1) as possible treatment option for wet Age-Related Macular Degeneration after topical application in mouse model

or it can also contain some main findings and then connect to AMD. Now it only tells that APNp1 role has investigated but nothing else. It is more like review article type title.

 Abstract

- In the abstract: ”We also tested delivery of APNp1 via injection of an adeno-associated virus (AAV) vector in a mouse model. ”

That should open little bit more also in abstract (but especially in the Methods part). How it was injected? Mention the route. Intravitreal injection? And how APNp1 is connected to AVV vector and administered? Now it remains open and this is not clear expression. Was APNp1 attached to AVV and then inject to eye so.? Or injected only together same time? Or wa APNp1 gene inserted into AVV vector and then injected? Should open short way but clearly to the reader.

- In the abstract:  ”…Finally, we investigated the mechanism by which APNp1 affects CNV by examining the expression of adiponectin receptor 1 (AdipoR1) and proliferating cell nuclear antigen (PCNA) in the choroidal tissue of mouse eyes. AdipoR1 and PCNA were overexpressed in mice with 24 laser-induced CNV compared to naïve mice.”

Please open little bit results from the study. There is nicely told what you studied next and then what kind of situation is pathological condition. there should also be mentioned how investigated treatment affects, the results of the present study.

- In the abstract: ” These data further our understanding of APNp1 as a therapeutic for wet-type AMD and present possible treatment alternatives that could be more beneficial for patients.”

Give some concretic exmaple beneficial findings based to the result. e.g. APNp1 showed potential to alleviate neovascularization upon wet AMD throug reducing….. Or something that based to results and then continue with sentence above (last sentence in the abstract). Now last sentence is not connected to the findings/results from the present study.

- Open in the abstract also FITC, fluorescein isothiocyanate

keyword: use words that have been used in the text not modified words e.g. Macular degeneration à age-related macular degeneration

 Introduction

- Introduction is quite short that should there be subheadings or just divided into different paragraphs? Maybe both are fine but came o mind.

- ”Progression of early AMD to late AMD (geographic atrophy and neovascular AMD) can have serious consequences, including vision loss and depression”

Give more relevant consequences related to AMD as difficulties to read or recognize faces etc. Suddenly depression sound little bit odd. Of course it could be there but explained little bit as secondary effect. As well before vision loss there is symptoms that affect to vision but not immediately lead to loss of vision even weaken it. Needs to open little bit more.

- ”There are treatments for wet AMD available”

Slowing treatments? Not cure, open little bit with few words.

- ”1.2 APN Peptide 1”

If abbreviations is mentioned first time in the subheading need to open there as well as in the first time mentioned in the text.

- Some part of the methods used need to explain more detailed and introduce how was AAV vector with APNp1 prepared and administrated. Latest in the Method section but as well shortly in abstract and may in introduction because there is mentioned present study related to AAV section.

”…AAV-delivered APNp1 to treat CNV”

 Materials and Methods

- Check that there is all product detailes (city and country).

- More animal information is needed e.g. animal number per group as well living conditions e-g- temperature, circadian rhythm, ad litbitum (?), habituation time, animal age at the testing time, some kind schedule need to show related to timetable of the administrattion, sacrification, tests for etc. All animal test permissions need to mention.

- in 2.1 ” APNp1 tagged with FITC and laser confocal microscopy were used to analyze cryosections of mouse eyes.”

Need to explain method more. Now it comes suddenly after information induce CNV. Reader not get message how and what there have been done. Administrated in this point APNp1 with FITC or labeled natural APNp1 in eye with FITCC? Need to open more.

-check that all abbreviations are also opened in the text when mentioned first time.

-ARPE-19 cells are mentioned first time in the Method part 2.2. It should come clear from the beginning also in the Abstract that there have used as well cell line with animal model.

- I have red text until 2.2 and be confused if there have investigated naivi mice and CNV induced mice until this point. From the abstract I thought that there is investigated topical eye drop effect. Explain more or make manuscript logical based to this and introduce all things investigated as in big picture.

-If all mice were treated as mentioned in 2.3 Move it upper after animal info or make more clear which animals were laser treated and which topical administered. Are those same or different animals. Now it is little bit confused.

- Add all product detailes related to purchase

- In section 2.4 procedure is nicely introduced and I miss same kind of introduce of procedures alreay from the beginning related to laser induced CNV and topical administration etc. Clear introduce of the all treatments and investigations.

Results

- Include bigger pictures that it is easier to detect results e.g. Figure 1 and Figure 2 A-F.

- Now there is results related to topical administration only in 3.2

After abstract I assumed that there is more related result to that. Please clarify abstract the way that there is mentioned different studies related to 3.1, 3.2 ja 3.3. It should come already clear in abstarct that basicly there is three different types procedures.

- Open figure 5 in the legend little bit more.

Discussion

-Discuss little bit more related to the previous literature. Now there is discussed mostly own results but not compared to the previous experiments. Now discussion basicly repeat results. More literature compared to the present study is needed.

Reviewer 3 Report

The authors group presented the anti-angiogenic effects of APNp1 in their previous manuscripts and in this study, they showed that topical and vector-used administration of APNp1 could induce APNp1 in the retina and choroid. Therefore, they insisted that topical or vector-used administration of APNp1 might suppress proliferation of CNV. This is a preliminary report of the author’s scenario and there were no validated data to suppress CNV proliferation.

Line 108. Please spell out PCNA. 

Figure 1. Since there is no detailed explanation on the method, I don’t understand this figure. First of all, what is CEC? Is it an endothelial cell of choroidal vasculature? Is this figure a flat mount of the choroid or cell culture of endothelial cells? Especially, figure 1B is unclear.

Line 138. Full spell of PCNA is unnecessary.

Figure 4. Fig.4A shows diffuse localization of APNp1 in the sensory retina, RPE, and choroid. Does this mean AdipoR1 distributes entirely in the retina and choroid?

Line 209. “both AdipoR1 and PCNA levels 210 were high in RPE cells affected by CNV.” Did AdipoR1 express in the proliferative RPE cells by laser photocoagulation or RPE cells apart from laser site?

I understand that proliferative cells express AdipoR1 and APNp1. Is the expression a cause of proliferation or effect of proliferation?

Round 2

Reviewer 1 Report

The authors have addressed most of my concerns. There are still some issues that need to be corrected.

1. I searched the APN primer sequences using NCBI blast software as well as DNA alignment software, however, it appears the forward and reverse primers of APN did not 100% match the APN sequences, while beta-actin primers 100% matched the  mouse beta actin gene sequences and generated a 103 base pair of PCR products. Please check the APN primer sequences carefully and correct the primer sequences, and also add which gene bank access number (for APN gene) was used for designing the specific primers.

2. From Figure 2G, it appears it is not the statistics result because there is no SD (standard deviation) bar,  please make sure to provide the statistical bar graph.

3. From line 141, please change sm2 to cm2

4. It appears the authors did not change "2x106 to 2X106

Reviewer 2 Report

Authors have revised manuscript and answered all my questions or suggestions. No more complaining. From my site it is ready for publication. 

Reviewer 3 Report

line 269-274. I don't admit the necessity of this paragraph, but if the authors want to describe it. This paragraph has to be revised. There are four commercially available anti-VEGF agents in many countries in 2022, ranibizumab, aflibercept, brolucizmab, and faricimab. Please describe latest information.
